# Trend and determinants of unmet need for family planning among married women in Ethiopia, evidence from Ethiopian demographic and health survey 2000–2016; multilevel analysis

**Meseret Desalegn Alemu** [ID]*, **Shimelash Bitew Workie** *, **Sintayehu Kussa, Tesfaye Tsegaye Gidey, Tezera Moshago Berheto**

School of Public Health, College of Health Sciences and Medicine, Wolaita Sodo University, Wolaita Sodo, Ethiopia

* meseretdesalegn58@gmail.com (MDA); sbitew0@gmail.com (SBW)

## Abstract

### Background

Unmet need refers to the gap between women's desires and contraception use to monitor their fertility level. According to the data, unplanned pregnancies are more likely to result in miscarriage and other obstetric difficulties, have poor maternal health care usage, and have a higher risk of having babies who are underweight. Information on the trend of unmet family planning needs in Ethiopia is scarce. The aim of this study was to examine the trend and determinants of unmet need for family planning among married or in union women in Ethiopia from 2000 to 2016.

### Method

Cross-sectional study design from secondary data was performed. Data for the study was obtained from four consecutive Ethiopian Demographic Health Surveys 2000 to 2016. The survey employs a nationally representative sample of households using a multistage stratified sampling technique. A descriptive analysis was done to see the trend in unmet need. Multivariable, multilevel logistic regression was performed to assess individual and community-level determinants. An adjusted odds ratio (AOR) at a 95% confidence interval and a p-value of 0.05 were used to declare the level of significance.

### Result

Unmet need declined by 40.2%, from 37.3% to 22.3%, from 2000 to 2016. Rural (AOR = 1.42; 95% CI: 1.27–1.59), number of living children > = 5 (AOR = 1.14 (1.04–1.24), age at first marriage > = 18 years (AOR = 1.15; 95% CI: 1.09–1.21), knowing at least one method of Family Planning (FP) (AOR = 1.57; 95% CI: 1.43–1.72), and no previous use of FP (1.27 (1.20–1.36) were associated with increased unmet need. While women between the ages of

**Data Availability Statement:** The datasets generated and/or analyzed during the current study (Ethiopia 2000, Ethiopia 2005, Ethiopia 2011 and

Ethiopia 2016) are publicly available from the Measure DHS website (https://dhsprogram.com/data/available-datasets.cfm?ctryid=65).

**Funding:** The author(s) received no specific funding for this work.

**Competing interests:** The authors have declared that no competing interests exist.

20 and 24 (AOR = 0.71; 95% CI: 0.64–0.79), 25–29 (AOR = 0.62; 95% CI: 0.55–0.70), 40–44 (AOR = 0.43; 95% CI: 0.39–0.50), 45–49 (AOR = 0.21; 95% CI: 0.18–0.24), the richest wealth index (AOR = 0.88; 95% CI: 0.80–0.96.

## Conclusion

The level of unmet need has decreased significantly in Ethiopia over the past 16 years. Age, educational level, media exposure, number of living children, age at first marriage, parity, previous use of FP, knowledge of FP, wealth index, regional setting, residence (rural), and survey year all have an association with an unmet need for family planning.

## Introduction

Unmet need for family planning refers to the proportion of women, who desire to stop or delay birth but are not taking any kind of contraception [1]. Unmet needs are classified as unmet need for limiting and unmet need for spacing. Unmet need for limiting refers to those women who are not currently using contraceptives, are not pregnant or amenorrheic but are able to child bear but want to delay their next birth for two or more years. While the unmet need for limiting are those women who are not currently on contraceptive use, not pregnant or amenorrheic but who are able to give birth but they do not want to give birth any more [2]. The common reasons for unmet needs overall are lack of knowledge, health concerns like side effects, misconceptions about contraception, perception of inability to conceive if contraception is stopped and husband's disapproval [3].

In 2019, out of 1.9 billion Women of Reproductive Age group (15–49 years) worldwide 1.1 billion have a need for family planning; of these, 270 million have an unmet need for contraception [1]. More than one out of every ten married or in-union women in the world has an unmet need for family planning [4]. Millions of sexually active women of reproductive age (15–49) in developing nations wish to avoid pregnancy and delay childbearing for at least two years. However, their needs for FP are unmet. Particularly in developing countries, unwanted pregnancies result in a series of problems on the health of woman and their families. Nearly 84% of unintended pregnancies in developing countries are related to an unmet need. Furthermore, it is estimated that there would be a three-quarter decline in unintended pregnancy in developing countries if the demand for family planning is well satisfied [5].

In Sub-Saharan Africa, the unmet need for family planning is greater than 20% [6]. Among the sub-Saharan sub-regions, Middle Africa has the highest unmet need for modern contraception at 31%, followed by Western Africa at 21%, Eastern at 19%, and Southern at 14% [7]. The unmet need for family planning becomes a health priority, which is addressed by Millennium development goals (goals 4 and 5) and Sustainable development goals (3.1 and 3.7) [8]. The goal aims to reduce maternal mortality to less than 70 maternal deaths per 100,000 live births and reduce the unmet needs to zero(0%) by 2030 [8]. The Health Developmental Army (HDA) and Health Extension Package is a key strategy, intended to scale up best practices including family planning. It is considered by the Ethiopian Health sector developmental plan (HSDP) -IV [9]. Evidences show that unintended pregnancies are associated with higher rates of miscarriage, and other obstetric complications, poor maternal health care utilization and increased risk of low birth weight [6, 10]. In resource-limited countries especially where sexual and reproductive health care infrastructures are poor, unintended pregnancy and abortion are significant barriers to the achievement of Millennium Developmental Goals (MDGs) related

to maternal and child health [10, 11] Young age pregnancies and birth are unintended in many settings. Each year, 12 million adolescents give birth and 3.2 million will face unsafe abortion [12]. Previous studies about determinants of unmet needs conducted in Ethiopia are fragmented in different settings. Many studies are without considering community-level factors or clustering effects. The finding of this study will provide the trend in unmet need based on national reliable data. In addition, it will reveal the clustering effect on unmet need. That is women from the same cluster will be similar than women from different cluster. There is a scarcity of information on the trend of unmet needs for family planning in Ethiopia. Therefore this study will used for evaluating the effectiveness of the programs by showing the trend and change in unmet need. It will be used as an input for health care providers, policy makers and programmers in which action must take to prevent the health problems due to unmet need through the identification of its determinants. In addition, it will used as baseline information for further studies. The aim of this study is to assess the trend and determinants of unmet need for family planning among married women in Ethiopia based on nationally representative data (EDHS 2000–2016). It will be used for program evaluation, as an input for health care providers and policymakers.

## Methods

### Source of data and setting

Cross-sectional study design was performed from secondary data from April to August 2022. The data for this analysis was obtained from four consecutive Ethiopian demographic and health surveys conducted in 2000, 2005, 2011, and 2016.

The EDHSs were conducted in all parts of Ethiopia in nine regional states and two administrative cities. All the Ethiopian demographic and health (EDHS 2000–2016) follow a cross-sectional study design [13–17]. All EDHSs follow a multistage sampling technique, using the sampling frame of a list of census enumeration areas (EAs) provided by the central statistical agency (CSA) conducted in 1994 and conducted in 2007 Population and Housing Census. In the first stage sampling of EA (clusters) was selected with systematic sampling with probability proportional to size. The household comprised the second stage of sampling. The detail is available in respective EDHS reports.

### Variables

**Dependent variable.**   The outcome variable is an unmet need for family planning, dichotomized as "unmet need" yes or "no." The total unmet need was calculated by adding the unmet need for spacing and the unmet need for limiting [18].

**Independent variables.**   *Individual-level factors (level 1)*. (Woman's age, Age at first marriage, Religion, Woman's and partners occupation, Exposure to media, number of living children, child mortality experience, Wealth index, visit a health facility, Knowledge at least one contraception, previous use of family planning, parity, Reason for not using, educational level of women)

*Community/cluster level factors (level 2)*. (Region, Place of residence).

### Operational definition

**Unmet need for family planning.**   The sum of the unmet need for limiting and unmet need for spacing is the total unmet need for family planning [19].

**Unmet need for spacing.**   includes pregnant women whose pregnancy was mistimed, postpartum ammenohoric whose last birth was mistimed and fecund women who are not

**Table 1. Definition of variables used in the study of trend and determinants of unmet need among married women in Ethiopia, EDHS 2000–20016.**

| Variable | Definition |
|----------|------------|
| Parity | is defined as the total number of birth a women give until the time of data collection |
| Media exposure | Media exposure is when a woman heard about FP from Radio, TV and newspaper or magazine |
| Previous use | If a woman previously used at least on of family planning methods |
| Reason for not using | Is the reason the respondent is not using a method of contraception to avoid pregnancy for those women who are not currently using a contraceptive method and who are not pregnant |

using any method of family planning, and say they want to wait two or more years for their next birth, undecided about the timing of next birth or undecided whether to have another children [19].

**Unmet need for limiting.** women who are pregnant whose pregnancy was mistimed, postpartum amenorrheic women whose last birth was mistimed and fecund women who are not using any method of family planning and say they don't want any more child [19].

Other variable definitioons used in this study are displayed in table below (Table 1).

**Data management and statistical analysis.** Data quality was assured for all EDHSs, and was available in respective EDHS reports [13–17]. Missing variables for >5% of cases were excluded from the model and cases missing for outcome variable were excluded. The data management and analysis were performed by STATA version 15. Data was extracted from IR (individual recode) file. The wealth index for the 2000 survey was constructed by using the statistical method of principal component analysis. Wealth status was then created from assets by placing households on a continuous measure of relative wealth after which households were grouped into five wealth quintiles namely poorest, poorer, middle, richer, and richest [19]. Items used to construct the wealth index were household owns a radio, television, electricity, kerosene lamp, bed/table, and electric mitad. And the type of flooring of the house, toilet facility, drinking water facility, number of members per room in the household, cattle, sheep, goats, house, and land were also used to construct the variable wealth index. Variable v005 is divided by one million (1,000,000) to generate population sample weight (wgt) to correct for over and under sampling and applied in all descriptive statistics.

Variables were recorded for the analysis. Exposure to media is categorized in to, have exposure to media and don't have exposure to media, which is generated from, reading about FP from newspaper, hearding about FP from radio and watching from television. The variable region is recoded into agrarian (Tigray, Amhara, Oromia, and SNNPR), urban (Addis Ababa, Dire dawa, and Harrari), and emerging/ pastoralist (Afar, Benshangul gumuz, Gambella, and Somali) for multilevel analysis. The data was declared as survey data with "svyset" command.

**Multilevel analysis.** In this study, two-level mixed-effects logistic regression analyses were employed using STATA software version 15. Since the EDHS data was hierarchical, i.e., women were nested in household and household were nested in cluster multi-level analysis is recommended. Because of the sampling approach used in the all EDHSs, women from the same cluster may be more similar to each other than women from the rest of the country. To account for this clustering, two-stage multivariable multilevel logistic regression analysis was used to estimate the effects of individual- and community-level determinants on unmet need and to estimate the between-cluster variability in the odds of unmet need. The datasets of EDHS 2000–2016 were merged together to assess the determinants of unmet needs. A multilevel logistic regression model in a combination of both fixed effect (a measure of association) and random effect (a measure of variation) was performed. Clusters were treated as random effects. The random effects were measured by the intra-class correlation coefficient (ICC),

median odds ratio (MOR), and proportional change in variance (PCV). The ICC shows the variation in unmet need for family planning for married reproductive women due to community characteristics.

$ICC = Va/\left(Va + \frac{\pi 2}{3}\right)$ Where, $V_a$ is area (cluster) level variance and $(\pi^2/3) \approx 3.29$ refers to the standard logistic distribution, that is, the assumed level-1 variance component [20].

The MOR was calculated by using the formula:

$$MOR = \sqrt{2*Va*0.6745}$$

The proportional change in variance was calculated as—PCV = $(V_A$-$V_B)/V_{A.}$

Where, $V_A$ is the variance of the initial model and $V_B$ is the variance in the subsequent models [20], [21].

Four models were fitted. Model I, with no determinants (random intercept) to estimate random variation in the intercept and ICC. Model II only included individual-level variables, model III only included community-level variables to estimate the community-level characteristics, and finally, model IV included both individual-level and community-level variables adjusted for both. The information criteria's Akaike Information Criteria (AIC) and Schwarz's Bayesian Information Criteria (BIC) were used to compare the models to choose the best fitted. The best-fit model was the model with the lowest AIC and BIC [22, 23]. A multilevel bivariable logistic regression model was employed and variables with p-value less than 0.25 were candidate variables for multilevel multivariable logistic regression model. The fixed effect size of individual and community-level factors using AOR at 95% CI was used to measure the association between outcome and determinant variables. A p-value of 0.05 was used to declare the significance. Multi-collinearity was checked by using variance inflation factor (VIF) and for each variable is less than 2.1 included in the model.

## Ethical consideration

The National Research Ethics Review Committee of Ethiopia (NRERC) and ICF Macro International approved all EDHSs. Permission from The DHS Program was obtained to use 2000–2016 EDHS data for further analysis after application with summary of proposal. This analysis was also approved by Ethical review Committee of College of health science and medicine; Wolaita Sodo University. Since this study was based on secondary data, participant informed consent was not applicable.

## Result

### Individual and community-level characteristics

In surveys conducted in 2000, 2005, 2010, and 2016 EDHS, 15,367, 14, 070, 16,515, and 15,683 women were interviewed respectively. From the total number of women interviewed 9,380, 8644, 10,204, and 9,824 women in 2000, 2005, 2011 and 2016 respectively were married or in a union [13–16]. Thus, those married/ in union women were included in the analysis. The majority, 87.8%% in 2000, 89.4%% in 2005, 82.1% in 2011, and 77.6% in 2016 were from rural areas and most were from the Oromia region, (38.5%, 36.4%, 38.5%, and 36.2%) in 2000, 2005, 2011 and 2016 respectively. The proportion of women with no education declined nearly by half from 83% in 2000 to 46.9% in 2016. While the proportion of those with primary and secondary education increased by threefold from 11.9% in 2000 to 34.9% in 2016 and from 5.1% in 2000 to 18.4% in 2016 respectively. The proportion of women whose partner with no education is decreased from 65.5% in 2000 to 45.8% in 2016 while those with primary education is increased from 23.1% in 2000 to 37.3% in 2016 (Table 2).

**Table 2. Weighted percentage of currently married/ in union women by individual and community-level factors, EDHS 2000–2016.**

| Background characteristics | | 2000 | 2005 | 2011 | 2016 |
|---|---|---|---|---|---|
| Number of women | Unweighted | 9,380 | 8,644 | 10,204 | 9,824 |
| | Weighted | 9,789 | 9,066 | 10,287 | 9,764 |
| Residence | Urban | (1,193) 12.2% | (960) 10.6% | (1,843) 17.9% | (2,192) 22.4% |
| | Rural | (8,596) 87.8% | (8,106) 89.4% | (8,444) 82.1% | (7,572) 77.6% |
| Region | Tigray | (627) 6.4% | (570) 6.3% | (620) 6.0% | (711) 7.3% |
| | Afar | (125) 1.3% | (109) 1.2% | (104) 1.0% | (86) 0.9% |
| | Amhara | (2,587) 26.4% | (2,330) 25.7% | (2,776) 27.0% | (2,245) 23.0% |
| | Oromia | (3,769) 38.5% | (3,300) 36.4% | (3,961) 38.5% | (3,34) 36.2% |
| | Somali | (112) 1.2% | (364) 4.01% | (232) 2.3% | (296) 3.0% |
| | Ben–gumuz | (111) 1.1% | (92) 1.0% | (124) 1.2% | (93) 1.0% |
| | SNNPR | (2,133)21.8% | (1,988) 22% | (2,021) 19.7% | (2,089) 21.4% |
| | Gambella | (30) 0.3% | (31) 0.4% | (41) 0.4% | (27) 0.3% |
| | Harari | (22) 0.2% | (22) 0.2% | (28) 0.3% | (24) 0.3% |
| | Addis | (236) 2.4% | (224) 2.5% | (342) 3.3% | (603) 6.2% |
| | Dire dawa | (38) 0.4% | (37) 0.4% | (38) 0.4% | (56) 0.6% |
| Educational level | No education | (8,121) 83.0% | (7,094) 78.2% | (6,735) 65.5% | (45,84) 46.9% |
| | Primary education | (1,161) 11.9% | (1,402) 15.5% | (2,861) 27.8% | (3407) 34.9% |
| | Secondary education | (455) 4.6% | (485) 5.4% | (378) 3.7% | (1189) 12.2% |
| | Higher education | (52) 0.5% | (85) 0.9% | (313) 3.0% | (584) 6.2% |
| Age | 15–19 | (862) 8.8% | (711) 7.8% | (765) 7.4% | (2,082) 21.3% |
| | 20–24 | (1,807) 18.5% | (1,574) 17.4% | (1,762) 17.1% | (1763) 18.0% |
| | 25–29 | (2,051) 21.0% | (2,065) 22.8% | (2,511) 24.4% | (1864) 19.1% |
| | 30–34 | (1,572) 16.1% | (1,551) 17.1% | (1,720) 16.7% | (1463) 15.0% |
| | 35–39 | (1,441) 14.7% | (1,343) 14.8% | (1,591) 15.5% | (1148) 11.8% |
| | 40–44 | (1,096) 11.2% | (960) 10.6% | (1,033) 10.0% | (808) 8.3% |
| | 45–49 | (961) 9.8% | (862) 9.5% | (905) 8.8% | (636) 6.5% |
| Parity | 0 | (896) 9.1% | (666) 7.4% | (916) 8.9% | (3.206) 32.8% |
| | 1–4 | (4,726) 48.3% | (4,466) 49.3% | (5,291) 51.4% | (3,863) 39.6% |
| | > = 5 | (4,167) 42.6% | (3,934) 43.4% | (4,080) 39.6% | (2694) 27.6% |
| Age at first marriage | <18 | (7,121) 72.7% | (6,292) 70.5% | (6,774) 65.8% | (5,837) 59.8% |
| | > = 18 | (2,668) 27.3% | (2,674) 29.5% | (3,513) 34.2% | (3,927) 40.2% |
| Media exposure | | N = 9,781 | N = 9,062 | N = 10,284 | N = 9,764 |
| | Yes | (1,482) 15.2% | (2,347) 25.9% | (3,494) 34.0% | (3,080) 31.6% |
| | No | (8,299) 84.8% | (6,715) 74.1% | (6,790) 66.0% | (6,684) 68.4% |
| Knows one of method | Yes | (8,334) 86.2% | (7,932) 87.5% | (10,038) 97.6% | (9,604) 98.4% |
| | No | (1,334) 13.8% | (1134) 12.5% | (249) 2.4% | (160) 1.6% |
| Previous use | Yes | (1,192) 12.2% | (2,184) 24.1% | (3,175) 30.9% | (3,626) 37.1% |
| | No | (8,597) 87.8% | (6,882) 75.9% | (7,112) 69.1% | (6,138) 62.9% |
| Wealth index | Poorest | (2,461) 25.1% | (1,759) 19.4% | (2,077) 20.2% | (1,633) 16.7% |
| | Poorer | (1,842) 18.8% | (1,892) 20.9% | (2,117) 20.6% | (1,721) 17.6% |
| | Middle | (2,072) 21.2% | (1,903) 21.0% | (2,083) 20.2% | (1,914) 19.6% |
| | Richer | (1,806) 18.4% | (1822) 20.1% | (1,923) 18.7% | (1,881) 19.3% |
| | Richest | (1,608) 16.4% | (1,689) 18.6% | (2,087) 20.3% | (2,614) 26.8% |

Most (21%, 22.8%, 24.4%, and 19.1%) of the respondents were from the age group of 25–29 years in 2000, 2005, 2011, and 2016 respectively. The proportion of women who got married at age of <18 years decreased from 72.7% in 2000 to 59.8% in 2016. Nearly 70.5% in 2005 and

65.8% in 2011 had gotten married before age of 18 years (Table 2). The majority, in 2000, 49.6%, in 2005, 47.1%, 43.7% in 2011, and 43.0% in 2016 were orthodox religious followers followed by Muslim religious followers (29.8% in 2000, 32.9% in 2005, 31.0% in 2011, and 20.2% in 2016) (Table 2).

## Level and trend of unmet need

The level of unmet needs was found 37.3% (95% CI: 35.7%-39.0%), 33.9% (95%CI: 32.5%-35.4%), 26.6% (25.0%-28.2%) and 22.3% (95% CI: 20.7%-23.9%). in 2000, 2005, 2011 and 2016 respectively. Over the past 16 years, the level of unmet needs declined by an average percentage of 0.94% per year from 2000 to 2016. The trend shows that unmet needs declined significantly by 40.2% from 2000 to 2016. The highest change in the proportion of women in unmet need is observed from 2005 to 2011 (reduced by 20.4%). While the unmet need for spacing decreased from 20.4% to 13.3% from 2000 to 2016. Whereas, the unmet need for spacing declined by nearly twofold from 16.9 in 2000 to 9% in 2016 (Fig 1).

The level of unmet need for family planning among women within the period of one-year post-partum was 31.2% in 2000, 33.0% in 2005, 24.7% in 2011 and 22.9% in 2016.

## Reason for not using

The questions are based on multiple responses, in which a woman can reply with more than one answer as a reason for not using family planning. The trend shows family planning non-user women due to fertility-related issues were found in 36.8%, 29.5%, 34%, and 48.4% in 2000, 2005, 2011, and 2016 respectively. The trend shows fear of side effects/health concerns to contraceptive use declined in four surveys, from 10.2% in 2000 to 5.4% in 2016. Lack of knowledge of FP methods is also mentioned as one of the reasons for not using contraceptives. Similarly, knows no method of family planning decreased from 7% in 2000 to 0.7% in 2016. The proportion of family planning non-users women due to religious prohibition ranges from 3.9% in 2005 to 5.9% in 2016. Lack of access cost too much and interference with the body was reported as reason for not using family planning (3.5% in 2000, 1.7% in 2005, 3.7% in 2011, and 2.8% in 2016) (Table 3).

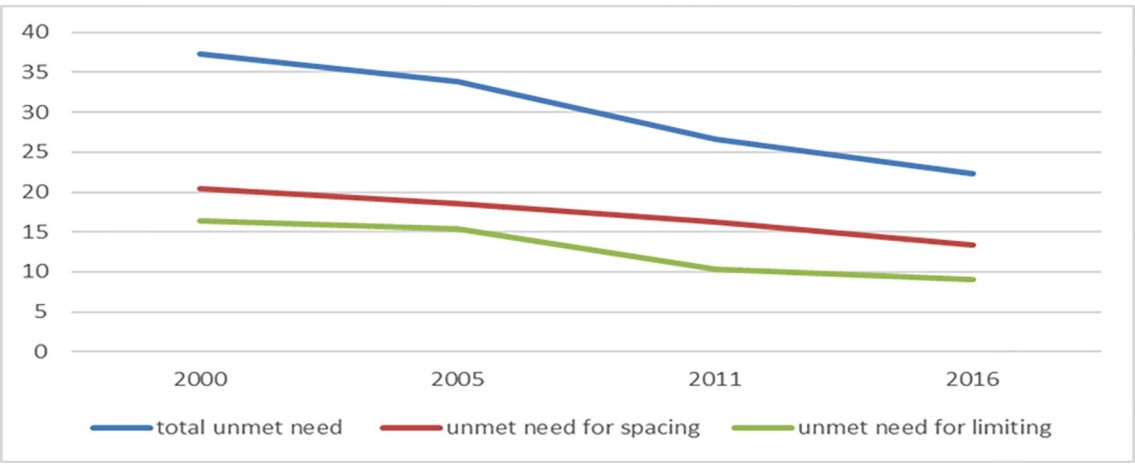

**Fig 1.  Trend of unmet need for spacing, unmet need for limiting and total unmet need among married/ in union women in Ethiopia, 2000–2016.**

**Table 3. Weighted percentage of married/ in union women by main reason for not using family planning, EDHS 2000–2016.**

| | Main reason | 2000 N | % of cases | % | 2005 N | % | %of cases | 2011 N | % | %of case | 2016 N | % | %of case |
|---|---|---|---|---|---|---|---|---|---|---|---|---|---|
| Fertility related | Not having sex | 483 | 11.1 | 8.2 | 508 | 9.3 | 11.6 | 858 | 14.6 | 18.6 | 114 | 6.2 | 6.6 |
| | Infrequent sex | 60 | 1.4 | 1.0 | 135 | 2.5 | 3.1 | 146 | 2.5 | 3.2 | 102 | 5.5 | 5.9 |
| | Menopausal/hysterectomy | 135 | 3.1 | 2.3 | 208 | 3.8 | 4.7 | 52 | 0.9 | 1.1 | 49 | 2.6 | 2.8 |
| | sub fecund/infecund | 155 | 3.6 | 2.7 | 77 | 1.4 | 1.8 | 61 | 1.0 | 1.3 | 82 | 4.4 | 4.7 |
| | Post-partum ammenohroic | 695 | 16.0 | 11.9 | 467 | 8.5 | 10.6 | 569 | 9.7 | 12.3 | 241 | 13.0 | 13.9 |
| | Breast feeding | 628 | 14.4 | 10.7 | 218 | 4.0 | 5.0 | 313 | 5.3 | 6.8 | 310 | 16.7 | 17.9 |
| | Total | | | 36.8 | | 29.5 | | | 34 | | | 48.4 | |
| Opposition | Respondent opposed | 234 | 5.4 | 4.0 | 130 | 2.7 | 3.4 | 162 | 2.8 | 3.6 | 135 | 7.3 | 7.8 |
| | Husband opposed | 146 | 3.3 | 2.4 | 149 | 2.7 | 3.4 | 152 | 2.6 | 3.3 | 77 | 4.2 | 4.5 |
| | Others opposed | 24 | 0.4 | 0.3 | 32 | 0.6 | 0.7 | 29 | 0.5 | 0.6 | 15 | 0.8 | 0.9 |
| | Religious prohibit | 319 | 7.3 | 5.4 | 213 | 3.9 | 4.9 | 322 | 5.5 | 7.0 | 110 | 5.9 | 6.3 |
| | | | | 12.1 | | 9.9 | | | 11.4 | | | 18.2 | |
| Lack knowledge | Knows no method | 409 | 9.4 | 7.0 | 440 | 8.0 | 10.0 | 130 | 2.2 | 2.8 | 13 | 0.7 | 0.7 |
| | Knows no source | 362 | 8.3 | 6.2 | 342 | 6.2 | 7.8 | 176 | 3.0 | 3.8 | 4 | 0.2 | 0.2 |
| | Total | | | 13.2 | | 14.3 | | | 5.2 | | | 7.2 | |
| Method related | Health concerns/ Fear of side effects | 598 | 13.8 | 10.2 | 565 | 10.2 | 12.8 | 402 | 6.8 | 8.7 | 100 | 5.4 | 5.8 |
| | Lack of access/Cost too much /Inconvenient to use/ Interferes with body | 204 | 4.7 | 3.5 | 134 | 1.7 | 2.1 | 217 | 3.7 | 4.7 | 52 | 2.8 | 3.0 |
| | Fatalistic | 466 | 10.7 | 7.9 | 263 | 4.8 | 6.0 | 277 | 4.7 | 6.0 | 304 | 16.4 | 17.6 |
| | Total | | | 21.6 | | 16.8 | | | 15.2 | | | 24.6 | |
| Other | | 899 | 21.4 | 15.9 | 1514 | 27.6 | 34.5 | 1963 | 33.4 | 42.5 | 136 | 7.4 | 7.9 |
| Don't k. | | 21 | 0.5 | 0.4 | 103 | 1.9 | 2.4 | 42 | 0.7 | 0.9 | 10 | 0.6 | 0.6 |
| Total | | | 134.9 | 100% | | 100% | 124.8% | 4614 | 100% | 127.2% | 1733 | 100% | 107.1% |

## Women's future intention to use family planning

In 2000 more than half (58.5%) of married women replied that they did not have the intention to use family planning in the future and the maximum proportion is recorded in 2000. This proportion decreased to 40.7% in 2005 and 43.5% in 2011. The trend in the proportion of women who are unsure about their future use of family planning is complex. It increased by one-fold from 3.2% in 2000 to 6.4% in 2005. However, this number decreased to 3.5% and 3.8% in 2011 and 2016 respectively (Fig 2).

## Factors associated with unmet need

A pooled weighted sample of 38,841 married women reproductive-age (15–19 years) women were included in the model.

**Random effects.** The empty model (the null model) revealed that the unmet need for family planning was not random across the communities (significant cluster level variance) (Va = 0.125, P < 0.001). The ICC was found to be 0.03 66. That is 3.66% of the chances of unmet needs being explained by cluster (community level) differences. The chi-square test in the intercept-only model is significant (p-value <0.01) which indicates the presence of a clustering effect in unmet needs. The between-cluster variability declined throughout the successive models from 3.66% in the random intercept model to 2.79% in model 2, 2.82% in model 3, and 2.65% in the final model (Table 4).

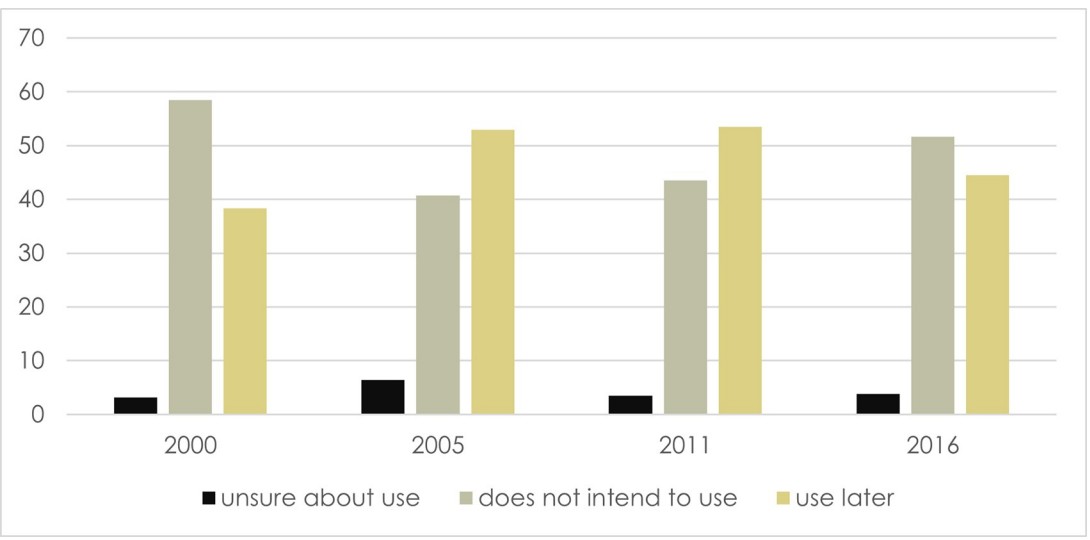

**Fig 2. Trend in percentage of currently married/ in union women (15–49) who are not using contraception and their future intentions regarding contraceptive use, Ethiopia 2000–2016.**

The full model, after adjusting for individual and community-level factors, about 28.8% of the odds of unmet need variation across communities was observed in the full model. The MOR for unmet need was 1.51 in the empty model, which indicated the presence of variation between communities (clustering) in unmet need since MOR was 1.51 times higher than the reference (MOR = 1). The unexplained community variation in unmet needs decreased to a MOR of 1.35 when all factors were added to the empty model (Table 4).

The odds of unmet needs among women who have 5 or more children is 14% (AOR = 1.14; 95% CI: 1.04–1.24) higher compared with those who don't have any living child. Women with parity 1 to 4 have a higher probability of unmet need than those with parity 0 (AOR = 2.57; 95% CI: 2.29–2.89). Unmet need is higher among women with a parity of more than 5 compared with women with a parity of 0 (AOR = 4.38; 95%CI: 3.82–5.01). The odds of unmet need were higher by 57% among women who know one method of family planning (AOR = 1.57; 95%CI: 1.43–1.72) compared to women who do not know any family planning method (Table 5).

**Table 4. Result from the random intercept model (measure of variation) of unmet need for family planning among married women at cluster level by multilevel logistic regression analysis.**

| random effects | Model 1 | Model 2 | Model 3 | Model 4 |
|---|---|---|---|---|
| Community-level variance/ *SE* | 0.125(0.134) | 0.095 (0.012) | 0.096 (0.012) | 0.089(0.012) |
| p-value | <0.001 | <0.001 | <0.001 | <0.001 |
| PCV | Reference | 24% | 23.2% | 28.8% |
| ICC (%) | 3.66 | 2.79 | 2.82 | 2.65 |
| MOR | 1.51 | 1.43 | 1.43 | 1.35 |
| AIC | 41,251.95 | 38,498.73 | 40,346.06 | 37,813.12 |
| BIC | 41,268.99 | 38,976.88 | 40,388.78 | 38,342.51 |

*SE-standard error, *ICC- intra-class correlation coefficient, *Model 1- empty model, a baseline model without any determinant variable, *Model 2—adjusted for individual level factors, *Model 3- adjusted for community-level factors, *Model 4- is adjusted for both individual and community-level factors

**Table 5. Individual and community-level determinants of unmet need among married women by using multivariable multilevel logistic regression model, EDHS 2000–2016.**

| Variables (reference: No education) | | Model 2, AOR(95% CI) | Model 3, AOR(95% CI) | Model 4, AOR(95% CI) | P–value (model-4) |
|---|---|---|---|---|---|
| Age(ref:45–49) | 15–19 | 4.92(4.2, 5.75)*** | | 4.86(4.15, 5.69) | <0.001 |
| | 20–24 | 3.63(3.18, 4.15)*** | | 3.45(3.01, 3.95) | <0.001 |
| | 25–29 | 3.09(2.73, 3.50)*** | | 3.01 (2.66 3.41) | <0.001 |
| | 30–34 | 2.42 (2.15, 2.73)*** | | 2.42 (2.14, 2.73) | <0.001 |
| | 35–39 | 2.15(1.91, 2.42)*** | | 2.16 (1.91, 2.43) | <0.001 |
| | 40–44 | 2.06(1.81, 2.33)*** | | 2.08 (1.84, 2.36) | <0.001 |
| Educational level | Primary education | 0.97(0.90, 1.04) | | 1.08 (1.01, 1.16) | 0.028 |
| | Secondary education | 0.76(0.67, 0.86)** | | 0.87(0.76, 0.99) | 0.030 |
| | Higher education | 0.53(0.41, 0.70) ** | | 0.75(0.57, 0.99) | 0.039 |
| Exposure to media (ref: no) | Yes | 0.86(0.81, 0.92) | | 0.86(0.80, 0.92) | <0.001 |
| Number of children (ref:0) | 1–4 | 1.02(0.94, 1.10) | | 1.03(0.95, 1.12) | 0.482 |
| | > = 5 | 1.09 (1.00, 1.19)** | | 1.14 (1.04, 1.24) | 0.011 |
| Parity (ref: 0) | 1–4 | 3.07 (2.11, 2.56)*** | | 2.57(2.29, 2.89) | <0.001 |
| | > = 5 | 5.39(4.72, 6.15)*** | | 4.38 (3.82, 5.01) | <0.001 |
| Previous use FP(ref: yes) | No | 1.41(1.32, 1.50)*** | | 1.27 (1.20, 1.36) | <0.001 |
| Knowledge FP (ref: no) | Yes | 1.54(1.42, 1.68)*** | | 1.57(1.43, 1.72) | <0.001 |
| Wealth index(ref:poorest) | Poorer | 1.15(1.07, 1.24)*** | | 1.03(0.95, 1.11) | 0.471 |
| | Middle | 1.20(1.11, 1.30)*** | | 1.05(0.97,1.14) | 0.195 |
| | Richer | 1.21 (1.12, 1.31)*** | | 1.06(0.98,1.15) | 0.145 |
| | Richest | 0.82(0.75, 0.90)*** | | 0.88(0.80, 0.96) | 0.006 |
| Age at first marriage(ref:<18) | > = 18 | 1.11(1.05, 1.17)*** | | 1.15(1.09, 1.21) | <0.001 |
| Residence (ref: urban) | Rural | | 2.25(2.08, 2.44)*** | 1.42(1.27, 1.59) | <0.001 |
| Regional setting(ref: agrarian) | Urban | | 0.79(0.72, 0.86)*** | 0.84(0.76, 0.92) | 0.002 |
| | Emerging | | 0.62(0.60, 0.68)*** | 0.70(0.66, 0.76) | <0.001 |
| Survey year(ref:2016) | 2000 | | | 2.82 (2.58, 3.09) | <0.001 |
| | 2005 | | | 2.24 (2.05, 2.44) | <0.001 |
| | 2011 | | | 1.87 (1.71, 2.04) | <0.001 |

Note: ref- reference, AOR- adjusted odds ratio ***- P-value < 0.001, **- p-value <0.05

The odds of unmet need are 4.86 times higher among women aged 15–19 than women aged 45-49(AOR = 4.86; 95%CI: 4.15–5.69). The odds of unmet needs among women aged 20–24 years were 3.45 times higher (AOR = 3.45; 95% CI: 3.01–3.95) compared to women aged 45–49. Unmet need was 3.01 times higher (AOR = 3.01; 95%CI: 2.66–3.41) among women aged 25–29 years and 2.42 times higher (AOR = 2.42; 95% CI: 2.14–2.73) among women aged 30–34 years (Table 5).

The odds of unmet need is lower by 13% (AOR = 0.87; 95% CI: 0.76–0.99) among women with secondary education and lower by 25% (AOR = 0.75; 95% CI: 0.57–0.99) among women with higher education compared to women with no education. Unmet need is less likely by 12% (AOR = 0.88; 95% CI: 0.80–0.96) among women from the richest household compared to women from the poorest household. Women who have exposure to media have lower odds of unmet need compared with women who do not have exposure to media (AOR = 0.86; 95%CI: 0.80–0.92) (Table 5). Women who got married at age of more than 18 years have a higher unmet need by 15% (AOR = 1.15; 95%CI: 1.09–1.21) compared to women who got married at age of less than 18 years. The odds of unmet need among women from the richest is 12% lower (AOR = 0.88 (95% CI: 0.80–0.96)) compared to women from the poorest household (Table 5).

Women from the rural areas have 42% higher unmet need compared to women from urban areas (AOR = 1.42; 95% CI: 1.27, 1.59). The odds of unmet need are 16% lower in urban regional settings (AOR = 0.84; 95%CI: 0.76–0.92) compared with the agrarian regional setting. The odds of unmet need is 2.82 times higher in 2000 (AOR = 2.82; 95%CI: 2.58–3.09), 2.24 times higher in 2011 (AOR = 2.24; 95% CI: 2.05–2.44), and 1.87times higher in 2011 (AOR = 1.87; 95% CI: 0.1.71–2.04) compared to 2016 (Table 5).

## Discussion

This study examined the individual- and community- level factors associated with unmet need for FP in Ethiopia and the trend in unmet need. Based on the empty model (the null model), unmet need for family planning was not random across the communities (explained community level variation). This finding is supported by the study conducted in Malawi [24]. This community level variation could be due to women from same cluster are more similar than women from different cluster. More than 3% of the chances of unmet need being explained by community level differences. This finding is in line with study done in Malawi which found 4% of unmet need is explained by community level variation [24]. One in five married or in-union women in Ethiopia had an unmet need for FP in 2016. The overall unmet need in sub-Saharan countries also ranges from 11.3% in Zimbabwe to 46.7% in Comoros [24]. However, the levels of unmet need in the first three surveys are higher than study in Nigeria (18%) [25]. Over the past 16 years, the level of unmet need declined by 40.2% from 2000 to 2016. The trend indicates that a significant decrease in unmet need was observed from 2000–2005, 2005–2011, 2011–2016, and 2000–2016 (non-overlapping CI). Based on this trend, it is difficult to attain the SDG goal of reducing zero (0%) unmet need by 2030 [8]. In line with this, the survey year is significantly associated with unmet need. As the survey year increases the odds of unmet need among women decreases. This decrement in unmet needs may be explained by an increase in the prevalence of contraceptive use from 6% in 2000 to 35% in 2016 [13,16]. An additional explanation may be the implementation of a health extension program (launched in 2002/2003 during HSDP II), initiation of youth-friendly services (YFS) in 2005 and increased health service accesses following additional health facilities being built [9,25]. Unequal decrement is observed across regions. The highest decrement was observed in the Amhara region (reduced by 59.2%) while the decrement is less than 10% in the Somali region. This regional variation in unmet need may be due to a difference in healthcare access and health service delivery.

There is a disparity between urban-rural residences. It might be due to urban women might have better access to health services, better initiation for using FP, and greater education and desire for more children in rural areas. However, this finding is lower than the study conducted in Dangla, SNNPR [26, 27]. The difference could be due to the difference in the sample size used.

Being from urban and emerging (pastoralist) regional settings is less likely to have unmet need compared to agrarian regional settings. The reason for unmet needs are less likely in the emerging region might be women from these regions do not want, or are unaware that they can, limit or space births, and also low contraceptive use in these areas. This could be explained by the difference in healthcare delivery distribution and access to health facilities and health facilities are more concentrated in urban areas. However, this finding is contrary to a study conducted in rural Ethiopia, which founds unmet need is higher likely in pastoralist regions [28].

Women who got their marriage at older age (>18 years) has higher odds of unmet need compared to women married at their younger age (<18years). This finding supported by study conducted in SNNP [26]. However, it is in contrary with the study conducted in Gonji Kolela [29].

Unmet need inversely related with women age. This finding is supported by a study conducted in India, Pakistan and Burkina Faso [28, 30, 31]. The odds of unmet need among women aged 20–24 years lower compared to women aged with 15–19 years. It confirms previous studies conducted in Mexico [32], sub-Saharan Africa [33], national level survey study in Ethiopia [34] and Enemay district [35]. This could be due to older women are matured enough and independent to decide their reproductive desire. Unmet need is higher among Muslim religious followers. This finding is consistent with a study conducted in Malawi and India [24, 36]. This could be due to religious prohibition for not using family planning. However it is in contrary to a study conducte in Nigeria, which founds unmet need, is less likely among Muslims [37].

Women who had no previous use of any family planning method were higher likely to have unmet need compared to women who had previous use of family planning. This finding is in line with a study conducted in Ghana and Tigray region [38, 39]. Unfortunately, the odds of unmet need are higher likely among women who know at least one method of family planning compared to women who did not know any method of family planning. A study conducted in Ghana also found that unmet need is more likely among women who know at least one method of family planning [40]. It is supported by a study conducted in SNNPR [27]. This might be due to women who don't know about FP may not know about the possibility of delaying or stopping childbearing and a lower need for FP. Even though it is contrary to expectation.

Women with secondary and higher education were less likely to have unmet need compared to women with no education. This finding is consistent with studies conducted in Ghana and Kenya [39, 41]. This finding is different from studies conducted in sub-Saharan Africa [24]. This might be due to educated women being more empowered to decide on contraceptive use, women who attained higher education women delay their marriage or childbearing age. However, unmet need is 8% more likely among women with primary education compared to women with no education. These results support the findings of previous studies of Klijzing [42] and studies conducted in Sub-Saharan Africa which found nearly 18% higher [24]. However, this finding is different from the study conducted in Mexico, and Pakistan [30, 32].

Unmet need is 14% higher likely among women with five or more living children compared to women with less than five living children. This finding is consistent with a study conducted in, Ethiopia, SNNPR [26, 43]. This might be due to women who have five or more children may achieved their number of children they desire. However it is lower than a study conducted in Kenya [41]. Women with a parity of 1 to 4 are more than two times more likely to have unmet need and more than four times higher among women with a party of five or more compared to women with a parity of 0. This finding collaborated with a study of Sub-Saharan Africa [24], a national level study in Ethiopia which found unmet need increased with an increase in parity [35]. However, It is different from a study conducted in India, which found unmet need is higher among women with parity zero and no significant difference with an increase in parity [44].

Being from the richest household is associated with decreased unmet need by 12%. This finding is supported by studies conducted in Pakistan, Mexico, and sub-Saharan Africa [24, 31, 33]. The possible reason could be richest women may have the freedom for decisions on their family planning and better access to family planning services. Women who have media exposure are lower likely to have unmet needs than those who do not have media exposure. This indicates information sharing on FP helps to address the barriers, which affect FP practice and to bring about a behavioral change.

## Strength and limitation

The strength of this study is the use of nationally representative standard data and a large sample size. Though the definition of unmet need was changed in 2012, this study is based on the former definition of unmet need and the difference in survey years is the limitation of the study. Since the study is based on secondary data; variables are restricted only to those available in the dataset. I.e. variables like quality of health services were not included in the analysis. Conclusion and recommendation.

The level of unmet need among married women in Ethiopia declined substantially over the study period, it is still unacceptably high. The highest proportion of unmet need was observed in rural and young age women. Age, age at first marriage, parity, knowledge of FP, residence, number of living children, Previous use of FP, wealth index, media exposure, educational level, and urban regional setting have associations with unmet need. Family planning policies should target rural and young women, regional specific family planning programs are needed. Moreover, qualitative research are needed to explore other reasons and barriers.

## Acknowledgments

We would like to say thanks to Wolaita Sodo University College of health sciences and medicine for facilitating this study. Special gratitude go to the DHS program office for giving permission to use the EDHS data freely.

## Author Contributions

**Conceptualization:** Meseret Desalegn Alemu, Shimelash Bitew Workie, Tezera Moshago Berheto.

**Data curation:** Meseret Desalegn Alemu.

**Formal analysis:** Meseret Desalegn Alemu, Shimelash Bitew Workie, Tezera Moshago Berheto.

**Methodology:** Meseret Desalegn Alemu, Shimelash Bitew Workie, Tezera Moshago Berheto.

**Software:** Meseret Desalegn Alemu, Shimelash Bitew Workie, Tezera Moshago Berheto.

**Validation:** Meseret Desalegn Alemu, Shimelash Bitew Workie, Tezera Moshago Berheto.

**Visualization:** Meseret Desalegn Alemu, Shimelash Bitew Workie, Tezera Moshago Berheto.

**Writing – original draft:** Meseret Desalegn Alemu, Shimelash Bitew Workie, Sintayehu Kussa, Tesfaye Tsegaye Gidey, Tezera Moshago Berheto.

**Writing – review & editing:** Meseret Desalegn Alemu, Shimelash Bitew Workie, Sintayehu Kussa, Tezera Moshago Berheto.

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
