## [Decision Letter · Decision Letter 0]

27 Jul 2023

PONE-D-22-33712Trend and determinants of unmet need for family planning among married women in Ethiopia, evidence from Ethiopian Demographic and Health Survey 2000- 2016; Multilevel AnalysisPLOS ONE

Dear Dr. Alemu,

Thank you for submitting your manuscript to PLOS ONE. After careful consideration, we feel that it has merit but does not fully meet PLOS ONE’s publication criteria as it currently stands. Therefore, we invite you to submit a revised version of the manuscript that addresses the points raised during the review process.

Major Revisions

We look forward to receiving your revised manuscript.

Kind regards,

Verda Salman, PhD

Academic Editor

PLOS ONE

Journal Requirements:

2. PLOS requires an ORCID iD for the corresponding author in Editorial Manager on papers submitted after December 6th, 2016. Please ensure that you have an ORCID iD and that it is validated in Editorial Manager. To do this, go to ‘Update my Information’ (in the upper left-hand corner of the main menu), and click on the Fetch/Validate link next to the ORCID field. This will take you to the ORCID site and allow you to create a new iD or authenticate a pre-existing iD in Editorial Manager. Please see the following video for instructions on linking an ORCID iD to your Editorial Manager account: https://www.youtube.com/watch?v=_xcclfuvtxQ.

Additional Editor Comments (if provided):

Major revisions required

Reviewers' comments:

Reviewer's Responses to Questions

**Comments to the Author**

1. Is the manuscript technically sound, and do the data support the conclusions?

Reviewer #1: Yes

2. Has the statistical analysis been performed appropriately and rigorously? 

Reviewer #1: No

3. Have the authors made all data underlying the findings in their manuscript fully available?

Reviewer #1: Yes

4. Is the manuscript presented in an intelligible fashion and written in standard English?

Reviewer #1: Yes

5. Review Comments to the Author

Reviewer #1: This is a good piece of research however following are the comments for further improvement of the manuscript.

1. Contribution of the study is not clear in the introduction section. Just saying that previous studies have not considered the clustering effects is not enough to claim the academic contribution of the study. What's the scholarly contribution of this study?

2. Definitions of all variables should be given in a table. There are few undefined variables e.g., "parity" is not defined and another ambigious variable is "reason for not using".

3. Wealth index is constructed by PCA and some discrete variables are used in this exercise, my question is regarding the use of correlation measure in PCA, is this the buil-in Pearson correlation measure or authors have used tetrachoric correlation? If Pearson is used then results are not valid for wealth index (see Islam, T. U., & Rizwan, M. (2020). Comparison of correlation measures for nominal data. Communications in Statistics-Simulation and Computation, 51(3), 698-714).

4. Mutilevel modelling: Give detailed econometric methodology with underlying distribution assumptions (if any). Also provide the diagnostic statistics to validate the model.

5. I could not find any discussion on the explained variation at the individual and community level.

6. Model 3 is unexplained.

6. PLOS authors have the option to publish the peer review history of their article (what does this mean?). If published, this will include your full peer review and any attached files.

Reviewer #1: **Yes: **Tanweer Ul Islam

---

## [Author Response · Author response to Decision Letter 0]

10 Sep 2023

Reviewer 1

1.Response: we have added the contribution of the study in the revised manuscript. 

2.Response: we have fixed it now.

3.Response: Thank you for recommending that interesting article and we referred it. for the construction of wealth index, variables were taken from Household Recode file. Discrete variables used for computing wealth index after recoded to a dichotomous variable/dummy variable based on the DHS standard criteria first and coded to 0/1. Then, biseral correlation was used.

4.Response: we have added a detail about the model in the revised manuscript.

5.Response: In the empty model, we simply aim to identify a possible contextual phenomenon that can be quantified by clustering of unmet need within neighborhoods. The ICC (intra-cluster correlation coefficient), MOR (median odds ratio), PCV(proportion change in variance) an the significant chi square test indicates the presence of clustering effect in unmet need.

Response: Model 3 is adjusted for only community level variables (as mentioned in the manuscript). However, the best-fitted model based on model selection criteria (AIC and BIC), model 4 is best fitted which was reported as final model.

---

## [Decision Letter · Decision Letter 1]

2 Oct 2023

PONE-D-22-33712R1Trend and determinants of unmet need for family planning among married women in Ethiopia, evidence from Ethiopian Demographic and Health Survey 2000- 2016; Multilevel AnalysisPLOS ONE

Dear Dr. Alemu,

Thank you for submitting your manuscript to PLOS ONE. After careful consideration, we feel that it has merit but does not fully meet PLOS ONE’s publication criteria as it currently stands. Therefore, we invite you to submit a revised version of the manuscript that addresses the points raised during the review process.

Major Revisions

We look forward to receiving your revised manuscript.

Kind regards,

Verda Salman, PhD

Academic Editor

PLOS ONE

Additional Editor Comments:

Major Revisions Required

Reviewers' comments:

Reviewer's Responses to Questions

**Comments to the Author**

1. If the authors have adequately addressed your comments raised in a previous round of review and you feel that this manuscript is now acceptable for publication, you may indicate that here to bypass the “Comments to the Author” section, enter your conflict of interest statement in the “Confidential to Editor” section, and submit your "Accept" recommendation.

Reviewer #1: (No Response)

Reviewer #2: All comments have been addressed

2. Is the manuscript technically sound, and do the data support the conclusions?

Reviewer #1: Partly

Reviewer #2: Partly

3. Has the statistical analysis been performed appropriately and rigorously? 

Reviewer #1: No

Reviewer #2: Yes

4. Have the authors made all data underlying the findings in their manuscript fully available?

Reviewer #1: Yes

Reviewer #2: Yes

5. Is the manuscript presented in an intelligible fashion and written in standard English?

Reviewer #1: Yes

Reviewer #2: Yes

6. Review Comments to the Author

Reviewer #1: My comments # 1, 4, & 5 are not incorporated and comment # 3 is partially incorporated.

. Contribution of the study is not clear in the introduction section. Just saying that previous studies have not considered the clustering effects is not enough to claim the academic contribution of the study. What's the scholarly contribution of this study?

2. Definitions of all variables should be given in a table. There are few undefined variables e.g., "parity" is not defined and another ambigious variable is "reason for not using".

3. Wealth index is constructed by PCA and some discrete variables are used in this exercise, my question is regarding the use of correlation measure in PCA, is this the buil-in Pearson correlation measure or authors have used tetrachoric correlation? If Pearson is used then results are not valid for wealth index (see Islam, T. U., & Rizwan, M. (2020). Comparison of correlation measures for nominal data. Communications in Statistics-Simulation and Computation, 51(3), 698-714).

4. Mutilevel modelling: Give detailed econometric methodology with underlying distribution assumptions (if any). Also provide the diagnostic statistics to validate the model.

5. I could not find any discussion on the explained variation at the individual and community level.

6. Model 3 is unexplained.

Reviewer #2: I suggest that unmet need according to different life-stages should be explored as well, particularly the period of one year postpartum.

7. PLOS authors have the option to publish the peer review history of their article (what does this mean?). If published, this will include your full peer review and any attached files.

Reviewer #1: **Yes: **Tanweer Ul Islam

Reviewer #2: **Yes: **Dr. Brinda Frey

---

## [Author Response · Author response to Decision Letter 1]

14 Nov 2023

Here are the responses for points raised

1. Contribution of the study is not clear in the introduction section. Just saying that previous studies have not considered the clustering effects is not enough to claim the academic contribution of the study. What's the scholarly contribution of this study?

 Response: we have added in the revised manuscript. 

2. Definitions of all variables should be given in a table. There are few undefined variables e.g., "parity" is not defined and another ambigious variable is "reason for not using".

 Response: we have fixed it now. 

3. Wealth index is constructed by PCA and some discrete variables are used in this exercise, my question is regarding the use of correlation measure in PCA, is this the buil-in Pearson correlation measure or authors have used tetrachoric correlation? If Pearson is used then results are not valid for wealth index (see Islam, T. U., & Rizwan, M. (2020). Comparison of correlation measures for nominal data. Communications in Statistics-Simulation and Computation, 51(3), 698-714).

Response: Thank you for recommending the article and we referred it. for the construction of wealth index, variables were taken from Household Recode file. Discrete variables used for computing wealth index after recoded to a dichotomous variable/dummy variable based on the DHS standard criteria first and coded to 0/1. Then, biseral correlation (one of tetrachoric correlation) was used.

4. Multilevel modelling: Give detailed econometric methodology with underlying distribution assumptions (if any). Also, provide the diagnostic statistics to validate the model. The model is constructed after checking the random effects

Response: we have added a detail about the multilevel model in the revised manuscript. Regarding econometric modelling, we do not have that. I could not find any discussion on the explained variation at the individual and community level. 

Response: In the empty model, we simply aim to identify a possible contextual phenomenon that can be quantified by clustering of unmet need within neighborhoods. The ICC (intra-cluster correlation coefficient), MOR (median odds ratio), PCV(proportion change in variance) an the significant chi square test indicates the clustering effect in unmet need. We used these results to check the whether multilevel model is required or not. However, we added in the discussion in the revised manuscript.

5. Model 3 is unexplained.

Response: Model 3 is adjusted for only community level variables (as mentioned in the manuscript). However, the best-fitted model based on model selection criteria (AIC and BIC), model 4 is best fitted which was reported as final model.

Reviewer #2: I suggest that unmet need according to different life-stages should be explored as well, particularly the period of one year postpartum

1. Thank you for your suggestion. We addressed in revised manuscript.

---

## [Decision Letter · Decision Letter 2]

12 Dec 2023

Trend and determinants of unmet need for family planning among married women in Ethiopia, evidence from Ethiopian Demographic and Health Survey 2000- 2016; Multilevel Analysis

PONE-D-22-33712R2

Dear Dr. Alemu,

We’re pleased to inform you that your manuscript has been judged scientifically suitable for publication and will be formally accepted for publication once it meets all outstanding technical requirements.

Kind regards,

Verda Salman, PhD

Academic Editor

PLOS ONE

Additional Editor Comments (optional):

Accepted for publication

Reviewers' comments:

Reviewer's Responses to Questions

**Comments to the Author**

1. If the authors have adequately addressed your comments raised in a previous round of review and you feel that this manuscript is now acceptable for publication, you may indicate that here to bypass the “Comments to the Author” section, enter your conflict of interest statement in the “Confidential to Editor” section, and submit your "Accept" recommendation.

Reviewer #1: All comments have been addressed

2. Is the manuscript technically sound, and do the data support the conclusions?

Reviewer #1: Yes

3. Has the statistical analysis been performed appropriately and rigorously? 

Reviewer #1: Yes

4. Have the authors made all data underlying the findings in their manuscript fully available?

Reviewer #1: Yes

5. Is the manuscript presented in an intelligible fashion and written in standard English?

Reviewer #1: Yes

6. Review Comments to the Author

Reviewer #1: I am satisfied with the following responses from the author(s). No more comments!

Here are the responses for points raised

1. Contribution of the study is not clear in the introduction section. Just saying that previous

studies have not considered the clustering effects is not enough to claim the academic

contribution of the study. What's the scholarly contribution of this study?

Response: we have added the contribution of the study in the revised manuscript.

2. Definitions of all variables should be given in a table. There are few undefined variables

e.g., "parity" is not defined and another ambigious variable is "reason for not using".

Response: we have fixed it now.

3. Wealth index is constructed by PCA and some discrete variables are used in this exercise,

my question is regarding the use of correlation measure in PCA, is this the buil-in Pearson

correlation measure or authors have used tetrachoric correlation? If Pearson is used then

results are not valid for wealth index (see Islam, T. U., & Rizwan, M. (2020). Comparison

of correlation measures for nominal data. Communications in Statistics-Simulation and

Computation, 51(3), 698-714).

Response: Thank you for recommending the article and we referred it. for the construction of

wealth index, variables were taken from Household Recode file. Discrete variables used for

computing wealth index after recoded to a dichotomous variable/dummy variable based on the

DHS standard criteria first and coded to 0/1. Then, biseral correlation was used.

4. Multilevel modelling: Give detailed econometric methodology with underlying

distribution assumptions (if any). Also, provide the diagnostic statistics to validate the

model. The model is constructed after checking the random effects

Response: we have added a detail about the model in the revised manuscript.

5. I could not find any discussion on the explained variation at the individual and community

level.

Response: In the empty model, we simply aim to identify a possible contextual phenomenon

that can be quantified by clustering of unmet need within neighborhoods. The ICC (intracluster

correlation coefficient), MOR (median odds ratio), PCV(proportion change in variance)

an the significant chi square test indicates the clustering effect in unmet need.

6. Model 3 is unexplained.

Response: Model 3 is adjusted for only community level variables (as mentioned in the

manuscript). However, the best-fitted model based on model selection criteria (AIC and

BIC), model 4 is best fitted which was reported as final model.

7. PLOS authors have the option to publish the peer review history of their article (what does this mean?). If published, this will include your full peer review and any attached files.

Reviewer #1: No

---

## [Editor Report · Acceptance letter]

21 Dec 2023

PONE-D-22-33712R2 

PLOS ONE

Dear Dr. Alemu, 

I'm pleased to inform you that your manuscript has been deemed suitable for publication in PLOS ONE. Congratulations! Your manuscript is now being handed over to our production team.

Kind regards, 

on behalf of

Dr. Verda Salman 

Academic Editor

PLOS ONE